# Ultrasound Localization Microscopy in Liquid Metal Flows

David Weik [1,*], Lars Grüter [1], Dirk Räbiger [2], Sanjay Singh [2], Tobias Vogt [2], Sven Eckert [2], Jürgen Czarske [1,3,*] and Lars Büttner [1,*]

1 Laboratory of Measurement and Sensor System Technniques, TU Dresden, 01069 Dresden, Germany; lars.grueter@tu-dresden.de
2 Department of Magnetohydrodynamics, Helmholtz-Zentrum Dresden-Rossendorf, 01314 Dresden, Germany; d.raebiger@hzdr.de (D.R.); s.singh@hzdr.de (S.S.); t.vogt@hzdr.de (T.V.); s.eckert@tu-dresden.de (S.E.)
3 Institute of Applied Physics, TU Dresden, 01069 Dresden, Germany
* Correspondence: david.weik@tu-dresden.de (D.W.); juergen.czarske@tu-dresden.de (J.C.); lars.buettner@tu-dresden.de (L.B.)

**Abstract:** Liquid metal convection plays an important role in natural and technical processes. In experimental studies, an instrumentation with a sub-millimeter spatial resolution is required in an opaque fluid to resolve the flow field near the boundary layer. Using ultrasound methods, the trade-off between the frequency and imaging depth of typical laboratory experiments limits the spatial resolution. Therefore, the method of ultrasound localization microscopy (ULM) was introduced in liquid metal experiments for the first time in this study. To isolate the intrinsic scattering particles, an adaptive nonlinear beamformer was applied. As a result, an average spatial resolution of 188 μm could be achieved, which corresponded to a fraction of the ultrasound wavelength of 0.28. A convection experiment was measured using ULM. Due to the increased spatial resolution, the high-velocity gradients and the recirculation areas of a liquid metal convection experiment could be observed for the first time. The presented technique paves the way for in-depth flow studies of convective turbulent liquid metal flows that are close to the boundary layer.

**Keywords:** ultrasound imaging; ultrasound localization microscopy; sub-diffraction imaging; ultrafast imaging; adaptive beamforming; magnetohydrodynamic convection

## 1. Introduction

### 1.1. Motivation

Convection is the cause of large-scale fluid motion in natural systems. It includes flow phenomena in oceans and the atmosphere. Convection in liquid metals is responsible for the dynamo effect in planetary cores [1,2] and plays an important role in technical systems, e.g., heat transport in solar thermal power plants [3,4]. Research on liquid metal convection is intensively concerned with formation of large coherent flow structures and scaling laws heat and momentum transport [5–7]. The simplified and fundamental configuration of Rayleigh–Bénard convection (RBC) is one of the classical hydrodynamic pattern paradigms for studying the properties of turbulent flows. In addition to the application of numerical methods, the studies rely on experiments using a low melting point metal alloy with a general and simplified configuration at laboratory scale [8–11]. However, the contribution and importance of these experiments depends on the available instrumentation. This is especially relevant for the study of large-scale flow structures (large-scale convection, LSC) [12–14]. Spatially resolved mapping of these regions would be very useful for the understanding of these flow phenomena. However, measuring the flow near the boundary layer in parameter regions of typical laboratory experiments requires sub-millimeter spatial resolution, which is not yet available.

### 1.2. State of the Art

Previous experimental studies have achieved breakthroughs by means of flow measurements using ultrasound methods. The potential of ultrasound for the flow mapping of liquid metals was first demonstrated by Takeda and Kikura [15]. The intrinsic oxides and impurities in the material act as scattering particles for the ultrasound [16]. Flow measurements in turbulent liquid metal convection are generally a challenge and so far, few experiments have been performed using the pulsed wave Doppler method (ultrasound Doppler velocimetry, UDV). It is a robust instrumentation, in which the estimated Doppler frequency shift is related to the flow velocity and the duration of flight to the position [17]. In our previous works, UDV was exploited for two-dimensional (2D) flow mapping [18] and for two velocity components (2D2C) using a combination of two ultrasound line arrays [19,20]. This method provided a frame rate of 10 Hz via the line scanning of the ultrasound transducers with time-division multiplexing and a spatial resolution of 5 mm.

Over the last decade, ultrasound scanning methods that are based on phased array plane wave imaging have evolved. Due to digital beamforming, phased array plane wave imaging enables 2D scans to be produced with image acquisition frequencies in the kilohertz range [21–23]. In a previous work, this method was combined with ultrasound image velocimetry (UIV) [24–26] and introduced for liquid metal flow mapping [27]. The spatial and temporal resolution was increased slightly compared to UDV. Moreover, one requisite for RBC experiments is to obtain a 2D2C velocity map using only a single access, which is met by the UIV method. The ultrasound localization microscopy (ULM) method has the potential to further increase the spatial resolution [28–31]. Single tracing particles are identified and tracked and then, assuming that there is an isolated particle, the spatial resolution is defined by how accurately the center of mass of that particle can be estimated [32]. As a result, the spatial resolution can be enhanced beyond the diffraction limit of ultrasound waves [33]. For high-resolution blood flow imaging, contrast agents are injected into the blood flow, which can be distinguished from other reverberating tissues by their harmonic echo frequencies [34]. When using the ULM approach, this method is called super-resolution ultrasound as the flow profiles can be measured, even in small vessels [35–37] and geometries [38].

For liquid metal imaging, however, contrast agents that overcome the high surface tension and that have a similar density to the liquid metal are not readily available. Therefore, the imaging has to rely on intrinsic scatterers only. A homogeneous distribution of these scattering particles throughout the entire volume is difficult to achieve due to sedimentation processes [16,39]. As a result, ULM was yet not been applied to liquid metal model experiments.

### 1.3. Aim and Outline

As a novelty in this work, we demonstrate how ULM can be applied for liquid metal flow mapping using nonlinear adaptive beamforming, as described in Section 2. In Section 3, the spatial resolution and uncertainty of ULM are discussed. It was shown that ULM can image beyond the diffraction limit, which is presented in Section 4 for the first time in a liquid metal RBC experiment.

## 2. Ultrasound Localization Microscopy

### 2.1. Ultrafast Imaging

The RBC experiment only allowed access from the side wall but required imaging for a penetration depth of at least 100 mm, supposing a symmetry of the flow field with respect to the center line of the convection cell. To optimize the focus in the elevation direction at increased penetration depths, a 1.5D phased array (Imasonic SAS, Voray sur l'Ognon, France) was used, along with the profile that is depicted in Figure 1a.

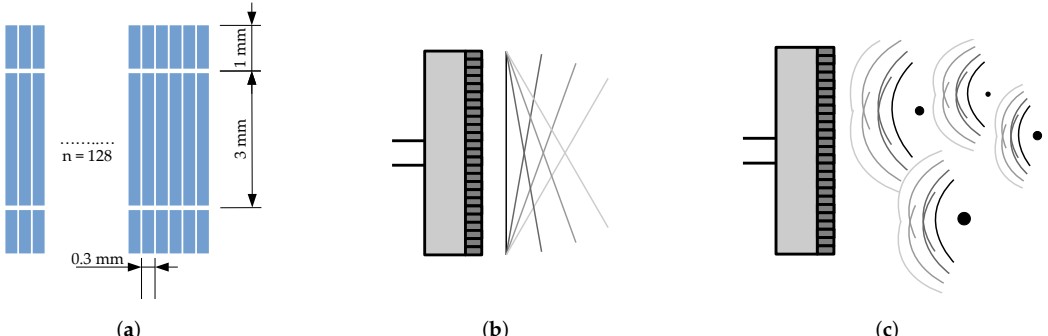

**Figure 1.** (**a**) The profile of the 1.5 D phased array transducer with inner and outer elements, a center frequency of 4 MHz and 70 % bandwidth; (**b**) the plane wave scanning with seven tilted plane waves of up to ±30° at a 1350 Hz *PRF*; (**c**) the receiving of backscattered echoes by the lower 96 inner elements.

The 128 inner and 128 outer elements were used in-phase for transceiving (TX) pulses and the 96 inner elements that were closest to the heating surface were used for receiving (RX). A scanning process that was based on ultrafast plane wave compounding [21] was used for the imaging, as shown in Figure 1b,c. Seven tilted plane waves reaching from −30° to 30° were excited with four sinusoidal periods at a center frequency of 4 MHz and a pulse repetition frequency (*PRF*) of 1350 Hz. This resulted in an ultrasound wavelength of $\lambda = 680 \, \mu m$ in GaInSn, which had a speed of sound of 2715 m/s. The inverse *PRF* 1/1350 Hz = 740 μs represented the time in which the incident pulse traveled five times back and forth through the 200 mm cube depth. This provided a sufficient attenuation of the acoustic energy to continue the imaging with the next plane wave. The attenuation was compensated by applying a constant digital time-dependent gain of 100 dB/ms. A research platform that was developed in-house (the phased array ultrasound Doppler velocimeter (PAUDV) [40]) was used for driving the ultrasound transducers and the analog signal amplification. The raw analog signals were digitized and stored on a disk by three 32-channel A/D converters (NI 5752B) and FPGAs (NI PXIe 7965, National Instruments, Austin, TX, USA) at a 20 MHz sampling rate.

*2.2. Nonlinear Beamforming*

To apply ULM, particles must be isolated from each other and from other disturbances. Conventionally, delay-and-sum (DAS) beamforming is applied to the raw ultrasound data in order to perform the re-localization of the echo information and, therefore, to image the scattering particles [41]. An example image is presented in Figure 2a. DAS produces too high a concentration of scattering particles and too high a speckle intensity. This is a result of the earlier mentioned scattering particle concentration that is not directly adjustable. The spatial resolution, especially for penetration depths of >50 mm, is not sufficient to isolate and track particles. To solve this problem, we proposed to apply a nonlinear beamformer. An approach that was based on coherence weighting was introduced by Matrone et al. [42]. The so-called filtered delay-multiply-and-sum beamformer (FDMAS) suppresses speckles and enhances the lateral resolution and coherent structures are emphasized by means of nonlinear inter-channel correlation. This method has been further enhanced by other research groups, such as the double-stage FDMAS beamformer [43]. However, these methods lack a dynamic scalability to particle densities and require a high computational effort to process that amount of data.

Polichetti et al. proposed the method of $p$th root compression delay-and-sum beamforming ($p$-DAS) [44]. The ultrasound signals of each tilt are compressed by their $p$th root, DAS beamformed and backward raised to the power of $p$. Afterward, each tilt is added. This method is a simplified version of FDMAS beamforming, but it produces comparable results, has a much higher computational efficiency and, most importantly, the grade of the nonlinear scaling is arbitrarily adjustable by the value $p$. This allows for the adaptive

scaling of the beamforming algorithm to the actual particle concentration. Hence, a $p$-DAS beamformer was implemented in OpenCL with GPU support in this study. To analyze its properties, the beamforming was conducted with several values of $p$. From Figure 2a, it can be seen that the particle profile sharpened as the value of $p$ increased. This also accounted for the ringing of the acrylic glass window that was visible for up to 20 mm. On the other hand, a loss of information in the corner regions and higher penetration depths could be seen for values of $p > 2$. To find the optimal value, particle tracking for each set of B-Modes was performed, as described in the next section. For $p = 2$, a compromise between the linked trajectories, the average trajectory length and the trajectories in the corner regions and higher penetration depths was found as the optimal value, as shown in Figure 2b. Furthermore, an apodization method that was based on two steps was applied during the beamforming. First, the sensitivity of each transducer element to each beamformed pixel was derived through the simulation and then normalized. A second apodization was calculated depending on the plane wave tilt angle. Pixels in the direct range were considered as "1" and pixels that were out of range were weighted with a Gaussian window that was based on the distance. Clutter filtering was applied using the coherent subtraction of the average beamformed time signals and background removal in the B-Mode data via PtvPy.

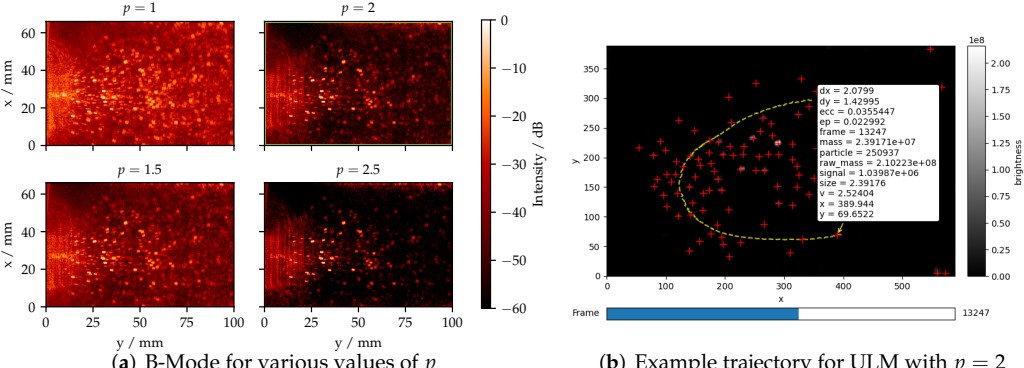

(**a**) B-Mode for various values of $p$        (**b**) Example trajectory for ULM with $p = 2$

**Figure 2.** Adjustments of the $p$-DAS beamformer to optimize localization and tracking using liquid metal echo data: (**a**) the data were beamformed with iterative values of $p$, where $p = 1$ represented conventional DAS beamforming; (**b**) the evaluation of particle tracking: $p = 2$ showed the best compromise between the number of identified trajectories (822 per minute), the average trajectory length (49.6 frames) and the identified trajectories in the corner regions.

### 2.3. Particle Localization and Tracking

For the localization and tracking of the scattering particles, the software package *PtvPy* [45], which was developed in-house, was used. *PtvPy* provides an integrated software tool that is based on the *Python* package *trackpy* [46], in which the localization and linking of particle positions is conducted according to Crocker and Grier [47]. First, the background image is calculated and subtracted from each frame. Second, the centroid method is applied for the localization of the center of each particle from the B-Mode images, which is commonly used in ultrasound imaging microscopy [48]. Third, the identified particles are linked to trajectories through the estimation of the most probable location between consecutive frames using the nearest neighbor method. Trajectories that could not be followed for at least 250 ms are neglected. A memory of three frames is used until a localized particle can vanish and reappear nearby to compensate for fluctuations due to noise. Finally, the mean velocity is calculated from the scattered trajectories using interpolation to a regularly spaced grid. This results in a two-dimensional and two-component (2D2C) flow velocity map.

### 3. Characterization of the Spatial Resolution

The spatial resolution of ULM is characterized by how well the center position of a single scatterer can be isolated. This also equals the localization uncertainty $\sigma_l$. In general, the localization uncertainty at a specific position $P(x, y)$ consists of a systematic uncertainty due to the finite size and ununiform shape of the particle profiles and image distortions, as well as a random uncertainty due to noise:

$$\sigma_l = \sqrt{\sigma_{l,\text{syst.}}^2 + \sigma_{l,\text{rand.}}^2}. \tag{1}$$

In accordance with the guide for the expression of uncertainty in measurement (GUM), the measurement uncertainty can be derived from a calibration measurement, in which the random uncertainty can be estimated using the standard deviation. The systematic uncertainty can be estimated using the average deviation from a reference. For ULM, this consists of two components in the $x$ and $y$ directions, through which the absolute localization deviation can be calculated using the root mean square:

$$\Delta l = \sqrt{(\Delta x)^2 + (\Delta y)^2}. \tag{2}$$

This is an unknown systematic deviation, which has to be considered with a uniform probability distribution as a worst case scenario. Therefore, this is translated into a Gaussian uncertainty by:

$$\sigma_{l,\text{syst.}} = \frac{\Delta l}{\sqrt{3}}. \tag{3}$$

As a reference, a set-up with a single scatterer was built [38]. It consisted of a fiber with a 50 µm diameter, which acted as scatterer. It was placed alongside the elevation direction of the ultrasound array inside a container. The container was designed to provide the same acoustic environment as was used in the demonstration experiment in Section 4 with an acrylic wall and GaInSn as the working fluid, as shown in Figure 3a. The fiber was moved through the container by two linear stages (LTM 80 and OWIS GmbH). The stages had a position deviation of 2.5 µm and a yaw angle deviation of 10 µm, which was negligible compared to the deviations of the ultrasound images. With this set-up, the point spread function was measured for several points on a defined grid inside the container. The center position was estimated using the same algorithm as ULM. The uncertainty $\sigma_l$ was calculated according to Equation (1) and is depicted in Figure 3b. The overall uncertainty was derived using the local average of all $N$ calibration points $P_N$ using:

$$\overline{\sigma}_l = \frac{1}{N} \sum_{n}^{N} \sigma_l(P_n) = 188 \, \mu\text{m} = 0.28 \, \lambda. \tag{4}$$

For comparison, the derived uncertainty was expressed relative to the wavelength. It can be seen in Figure 3b that the uncertainty was not dependent on the penetration depth, as long as a sufficient signal-to-noise ratio, or rather peak-to-background ratio, was present and the diverging spatial resolution was sufficient to separate particles. This was also an issue in shadowed regions or away from the directivity of the phased array. We were able to extract calibration points at a distance of 100 µm from the side wall, which was supposedly within the region of the viscous boundary layer in the experiment (<0.5 cm).

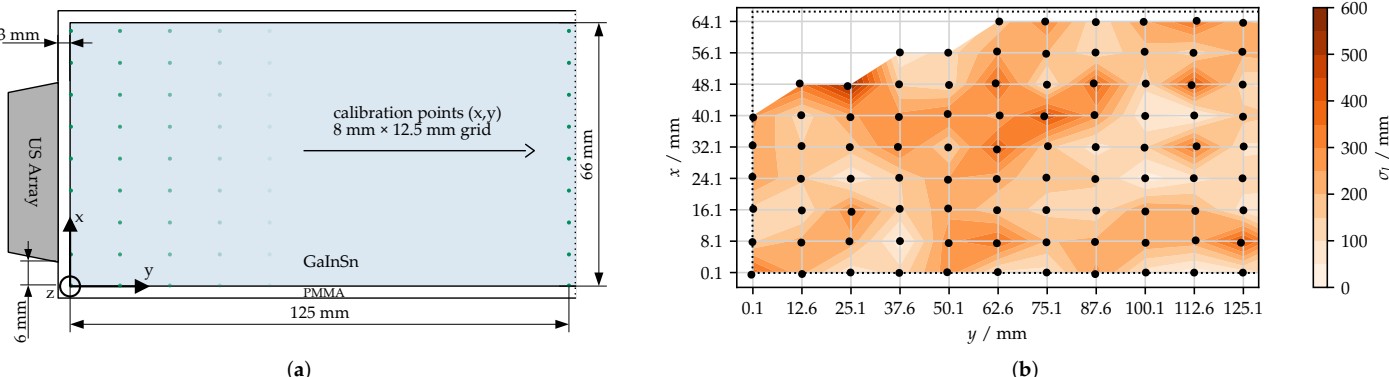

(**a**)                                                                                              (**b**)

**Figure 3.** Calibration of the localization uncertainty: (**a**) the set-up, in which the experiment dimensions and materials were replicated and a fiber that used as the scatterer was positioned inside the container; (**b**) the calibration result, in which the position of the scatterer was estimated and the uncertainty was derived using the reference position. The overall average was $\overline{\sigma}_l = 188\,\mu m$.

## 4. Test Case: Rayleigh–Bénard Convection Experiment

### 4.1. Set-Up

The convection cell for this study was designed at the magnetohydrodynamics lab at Helmholtz-Zentrum Dresden-Rossendorf (HZDR) [8]. The container had a square cross-section with a side length of $L = 200\,mm$ and a height of $H = 66.6\,mm$. This produced an aspect ratio of $\Gamma = {}^L/_H = 3$. The fluid layer was located between two copper plates on the top and bottom of the cell, in which there was a network of channels that were connected to a temperature-controlled water circulation system. The strength of the thermal forcing is described using the dimensionless Rayleigh number $R_a$. The thermal boundary conditions were controlled by thermostatic baths and the temperature gradient across the liquid metal layer was controlled by nine thermocouples in each plate. The side walls were made of electrical non-conductive polyvinyl chloride. The influence of heat loss and environmental temperature conditions was reduced by encasing the entire convection cell in 30 mm insulating foam. The fluid vessel was filled with the eutectic alloy GaInSn as the model fluid. At room temperature, it had a density of $6.44\,g/cm^3$, a viscosity that was approx. three times lower than water $3.4 \times 10^{-7}\,m^2/s$ and a thermal diffusivity of $1.05 \times 10^{-5}\,m^2/s$. Accordingly, the Prandtl number was 0.03. The convection cell was placed between two electromagnetic coils to provide a horizontal magnetic field. The spatial inhomogeneity of the magnetic field was less than 5% at maximum strength [49]. Different flow patterns appeared in the liquid metal convection depending on the ratio between the Lorentz force and the viscosity, which was represented by the Chandrasekhar number $Q$. The ultrasound array probe was placed upright with 66 mm offset from the side. The measurement area of the probe was aligned perpendicularly to the magnetic field lines, as shown in Figure 4.

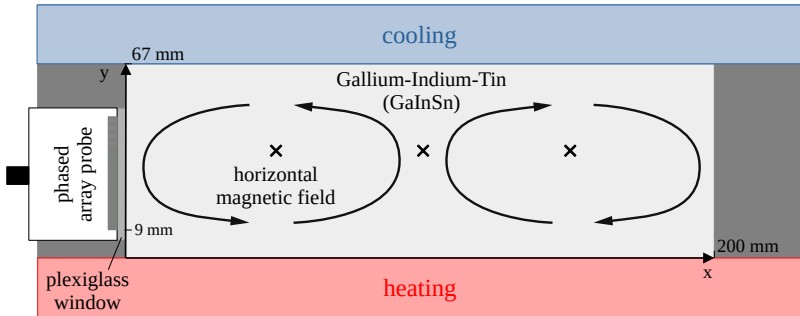

**Figure 4.** Cross-sectional drawing of the experimental set-up in which the ultrasound probe was situated. The probe was attached to the medium with a thin film of high-temperature couplant agent at an acrylic glass window that had a similar speed of sound as GaInSn. The horizontal magnetic field was aligned perpendicularly to the measurement plane and caused the turbulent convection to organize itself into ordered convection rolls.

### 4.2. Application

To demonstrate the potential of ULM, an experiment was conducted using the RBC set-up at HZDR. The experiment was performed with a 2 K and 10 K temperature gradient, which corresponded to $R_a = 2 \times 10^5$ and $R_a = 10 \times 10^5$, respectively. The application of a fairly strong magnetic field of 300 mT, which corresponded to $Q = 2 \times 10^5$, resulted in a relaminarization of the flow. Both experiments were recorded for 10 min by ULM. The time-averaged flow structures are depicted in Figure 5.

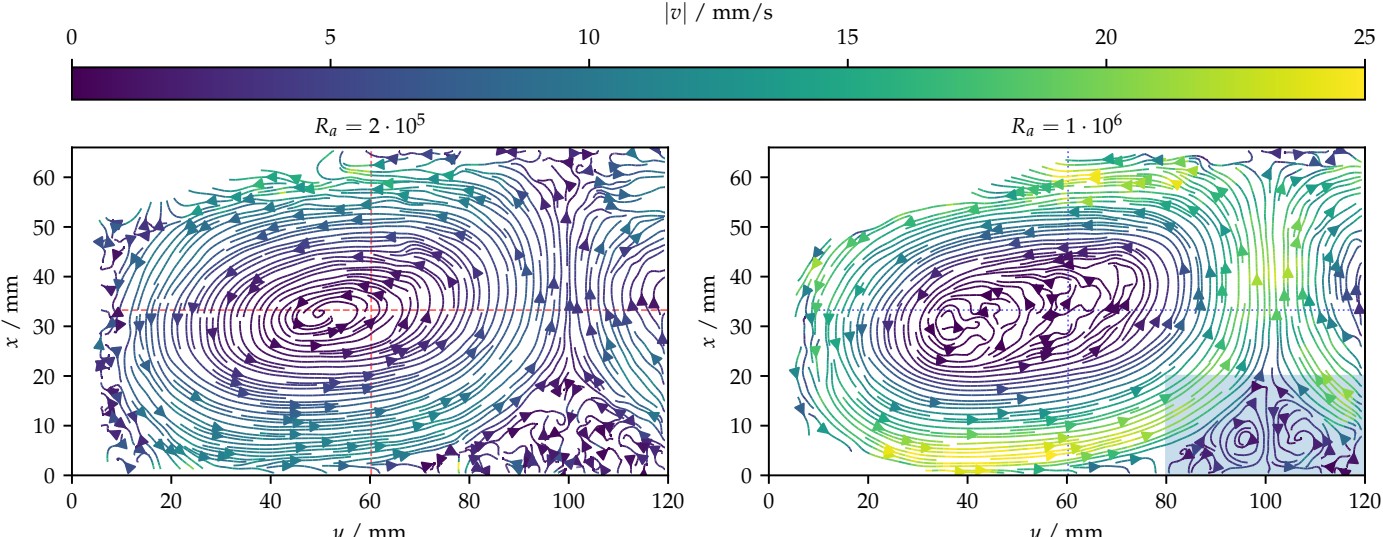

**Figure 5.** Measured streamlines averaged over 10 min of the experiment with (**left**) $R_a = 2 \times 10^5$ and (**right**) $10 \times 10^5$, for which the indicated cross-sections are shown in Figure 6 and the zoomed region is shown in Figure 7.

It was shown that both flow regimes consisted of two convection rolls with an elliptic cross-section, in which the first and, partly, the second were resolved with a maximum penetration depth of 120 mm. Due to the acoustic directivity pattern of the ultrasound probe, the upper left-hand part and the small lower left-hand part were shadowed. The upper shadowed region increased due to the lower displacement of the ultrasound probe. As expected, the experiment with the lower $R_a$ incorporated a decreased velocity range. The recirculation areas were not as distinct as in $R_a = 10 \times 10^5$. The velocity distribution of the main roll in $R_a = 2 \times 10^5$ appeared to be more regular with a decreased velocity gradient and circular streamlines almost into the center. This can also be seen in the cross-

sections of the velocity in Figure 6. ULM showed that the zero velocity region was larger with an increased $R_a$. This indicated the transition into turbulent flow convection.

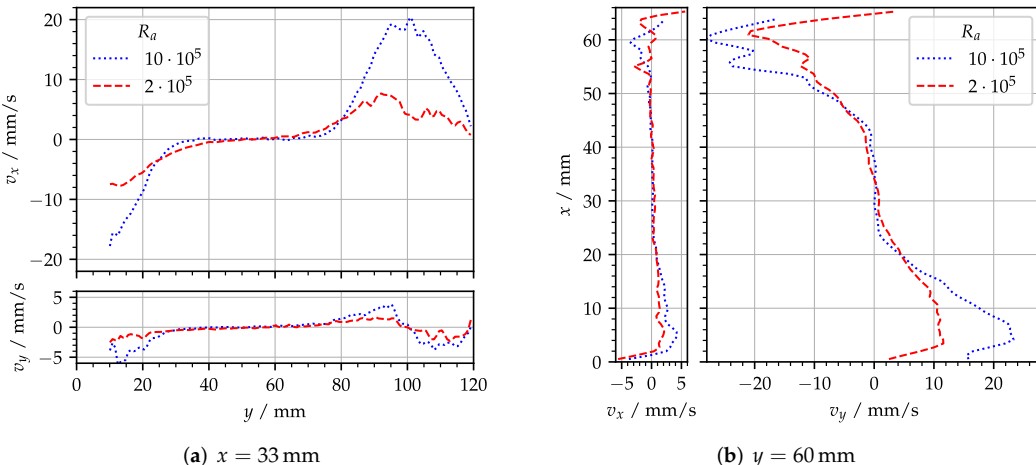

**Figure 6.** Cross-sections of both experiments in the (**a**) y direction and (**b**) x direction with both velocity components.

Furthermore, an enlarged section of the recirculation area with $R_a = 10 \times 10^5$ is shown in Figure 7.

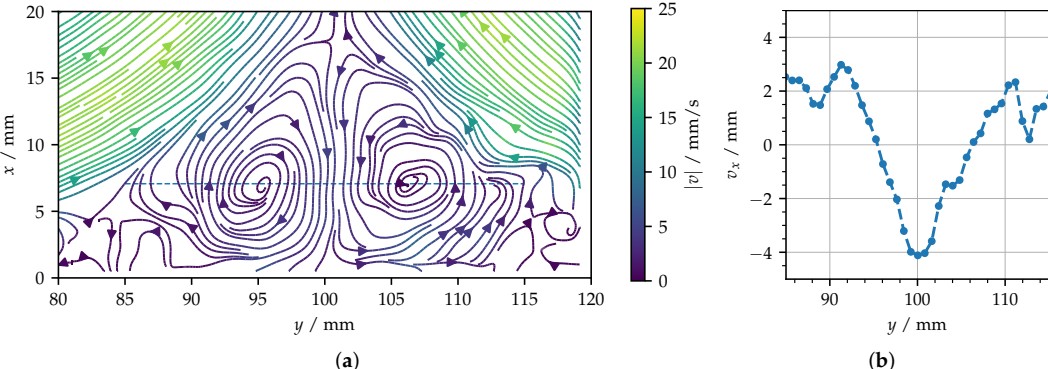

**Figure 7.** Enlarged recirculation area with $R_a = 10 \times 10^5$, shown as (**a**) the streamlines and (**b**) the x component of the velocity vector along the cross-section in the y direction.

To our knowledge, this result was the first spatially resolved recirculation area with counter-rotating rolls to be taken from an liquid metal RBC experiment. Here, the sub-millimeter spatial resolution was demonstrated as the dimensions, flow gradients and the shape of the recirculation rolls were obtained.

Although the time-averaged results revealed a clear and stable flow structure, the actual flow was also subject to significant temporal fluctuations. The temporal dynamics could be obtained from the time-resolved tracking data. An animation of a 1 min measurement period is attached as Supplementary Material to this paper. It was found that the inner regions of the main convection roll were rotating in tiny swirls with a radius of 1 mm to 2 mm in certain patches, whereas the time-averaged data show almost no velocity in that region. On the contrary, the main flow of the roll followed more static paths. Furthermore, a high-velocity gradient in the convection roll could be obtained from close trajectories. This was possible due to the high spatial resolution that was achieved with ULM.

## 5. Conclusions

In this work, a flow velocity measurement method using ULM was presented for environments that do not allow for the insertion of artificial tracing particles but do require a high spatial resolution. It enabled flow studies with an average spatial resolution of 188 μm at an excitation wavelength of 680 μm. The comparable low–mid frequency and the 1.5D phased array probe allowed for a broad imaging area of 66 mm in the lateral direction and a penetration depth of up to 120 mm.

This is relevant in liquid metal flow studies that require a high spatio-temporal resolution, for instance. For those conditions, an adaptive beamforming approach that was based plane wave compounding and nonlinear beamforming was applied. The beamformer could be scaled to dynamically filter the particle information in order to isolate the particles, which solved the pre-condition of the localization microscopy approach. This method is furthermore important for biomedical blood flow measurements, as the concentration of intravenously injected contrast agents also changes dynamically. As a test case, ULM was demonstrated in an RBC experiment, in which the flow map of the recirculation area in between convection rolls could be observed for the first time in detail. The high flow velocity gradients could be analyzed directly through the experimental studies. Despite this, the measurement of velocity profiles within the viscous boundary layer would require an even higher spatial resolution and further research efforts within the field of ULM. This could be conducted by means of higher ultrasound frequencies, the combination of further distributed ultrasound transducers, advanced beamforming [50,51] or linking algorithms [52,53]. Nonetheless, the experimentally obtained ULM data revealed flow details for laboratory liquid metal experiments that had not be obtained so far and thus, allowed us to close the gap between theory, numerics and experiments.

**Supplementary Materials:** The following supporting information can be downloaded at: https://www.mdpi.com/article/10.3390/app12094517/s1.

**Author Contributions:** Conceptualization, D.W., L.G., L.B., T.V., S.E. and J.C.; methodology, D.W. and L.G.; software, D.W. and L.G.; investigation, D.W. and S.S.; resources, D.R., S.S., T.V. and S.E.; data curation, S.S., D.W. and L.G.; writing—original draft preparation, D.W.; writing—review and editing, D.W., L.B., S.E., T.V. and J.C.; visualization, D.W. and L.G.; supervision, L.B., J.C., T.V. and S.E.; project administration, L.B., J.C., T.V. and S.E.; funding acquisition, L.B., J.C., T.V. and S.E. All authors have read and agreed to the published version of the manuscript.

**Funding:** This research was funded by the Deutsche Forschungsgemeinschaft (DFG), grant numbers BU 2241/2-2, CZ 55/43-1, EC 217/2-2, VO 2331/1-1 and VO 2331/4-1.

**Institutional Review Board Statement:** Not applicable.

**Informed Consent Statement:** Not applicable.

**Data Availability Statement:** The data presented in this study are available on request from the corresponding author.

**Acknowledgments:** We would like to further acknowledge Christoph Schöne for his assistance with data analysis, Julius Weber for his work on the calibration and the inspiring ideas and fruitful discussion from former members of the ultrasound imaging group, namely Richard Nauber and Christian Kupsch.

**Conflicts of Interest:** The authors declare no conflict of interest. The funders had no role in the design of the study, the collection, analyses or interpretation of the data, the writing of the manuscript or the decision to publish the results.

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
