# Peer review of "Ultrasound Localization Microscopy in Liquid Metal Flows"

_applsci, doi:10.3390/app12094517_

Round 1

Reviewer 1 Report

The manuscript is easy to follow and clear. The use of ULM for investigating the flows in liquid metals is interesting. This manuscript presents the first experimental study on this aspect. Although it makes this work experimentally original, it lacks substance from a fluid dynamics point of view (one-case study) and contains methodological errors from an ultrasound point of view. Additional flow data should be reported, and the ultrasound acquisitions should be improved.

1) A Rayleigh-Bénard convective (RBC) flow was investigated. A single case was investigated. The results do not allow concluding i) if the technique works for other RBC flows, ii) if the estimated velocities and fields are correct:

  1. i) Only one velocity field is presented in the manuscript. Several flow configurations, i.e. several Rayleigh numbers Ra, must be analyzed. This will ensure that the proposed technique works in several situations. As the article is presented, one gets the misperception that the authors have presented the only case that works.
  2. ii) Comparisons against CFD must be done (2-D RBC flow simulations should be suitable). The velocity fields measured by ULM (with different Ra numbers) could thus be compared (at least qualitatively) with those simulated by CFD.

2) Ultrasound images were created through compounding by transmitting series of seven tilted planes waves. The tilt angles were as large as 30 degrees. With a center frequency of 4 MHz, and a pitch of ~0.3 mm, such large angles generate strong grating lobes (amplitude as large as 20 dB). This counter-intuitive approach must be clearly motivated because the transmission conditions were not optimal. I would recommend angles no greater than 15 degrees.

3) I do not understand how the PRF was defined (“five times back and forth”).

4) Please clarify the sentence #118 regarding apodization, tilt angle, and sensitivity.

5) I would recommend using a clutter filter to mitigate the ringing artifacts.

6) Please clarify Equation (2).

Author Response

1) The RBC configuration was chosen to demonstrate the suitability of the measurement method. The chosen configuration differs only slightly in aspect ratio from the previously extensively studied liquid metal RBC in gamma=5 [1-4]. The Technique is particularly relevant for non-transparent liquid metal experiments. The transferability to other liquid metal experiments than the ones presented here depends mainly on the temperature range and velocity range.

1. i) RBC flows at different Ra numbers differ essentially in the resulting velocities. For the minimum velocity of ULM, the localization uncertainty and the length of the observed trajectory are relevant. But slower particles can usually be observed for a longer time. For the maximum velocity, tracking errors are limiting, if the particle movement is not distinguishable from others. With the typical velocity gradients, the particle densities and the localization uncertainty a rough estimation for the velocity range with the shown ULM instrumentation can be given by 0.05 mm/s < v < 80 mm/s. We further added the result of a second RBC experiment with a lower Ra number, to show more experimental results in this paper.

1. ii) The aim of this paper is to demonstrate the functionality of the measurement principle for liquid metal experiments. In the future, more works with a focus on fluid dynamics are planned. Direct numerical simulations for liquid metal convection that provide the required accuracy are extremely costly because extremely small Kolmogorov scales must be resolved compared to fluids such as water or air due to the low Prandtl number. A 2D CFD simulation would not be appropriate for this purpose.
We suggested that the plausibility of the results is provided by the calibrated localization uncertainty. With the pre-condition of isolatable scatterers, the velocity uncertainty can be derived from the static localization uncertainty, if flow specifics such as the velocity gradient or the length of the followed trajectory are modelled with error propagation [5].

2) The angle of 30° was deliberately chosen to improve the imaging in the corner regions next to the ultrasound probe. It is more or less based on an optimization conducted in earlier studies with the exact same probe in a similar environment [6]. Nonetheless, grating lobes with these amplitudes would introduce quite disturbing artifacts, where reducing the angles is an interesting approach. However, it might be the case that intuition coming from typical ultrasound applications in biological tissues is a bit misleading in liquid metals. In medical imaging, the speed of sound usually ranges from 1.5 to 1.6 km/s, whereas in GaInSn in the range of 2.7 km/s. With 4 MHz, this leads to a wavelength of 0.68 mm in our setup. Therefore, no grating lobes should appear, as the half-wavelength-criteria is fulfilled. Furthermore, we simulated the radiation of a 30° tilted wave with the ultrasound probe in the environment (3 mm plexiglass window into GaInsn) using an implementation of the EFIT2D algorithm. As a result, the simulation reveals no disturbing grating lobes using these parameters.
However, thanks to your comment we recognized, that the speed of sound in GaInSn was not mentioned in the manuscript. As this can be misleading, we added this remark in Sec. 2.2, where the wavelength was described: 
“This results in an ultrasound wavelength λ = 680 µm in GaInSn, which has a speed of sound of 2715 m/s.”

3) It is defined as to the time, when a pulsed plane wave has travelled five times through the cube with a length of 200 mm. Which means 2715 m/s / (5 * 2 * 0.2 m) with c = 2715 m/s. For clarity, we changed this sentence to:
“The inverse PRF 1 / 1350 Hz = 740 µs represents the time, when the incident pulse has travelled five times back and forth through the 200 mm cube depth.”

4) Thank you for your remark. We did not elaborate on the apodization as to enhance the focus or scope of the paper. However, if you in truth are correct with comment 2), this would be a very crucial part in order to deal with the grating lobes. Due to your comment, we added a few more explanations in the manuscript, but as we do not think that grating lobes are an issue, we suggest this not as a main topic of the paper and kept it short. It is likely that any other apodization function gets similar or even better results. Anyhow, a thorough explanation on the apodization is given here:
The used apodization is based on two steps. First the sensitivity of each transducer element to each beamformed gate is derived by acoustic simulations. This is done with the simulation of a point source at each beamformed gate: a pulse is radiated, the echo is received with the ultrasound probe and the echo intensity of each RX channel is derived. The apodization of each channel is then calculated with the echo intensity divided by the maximum echo intensity. This can also be referred to a time reversal apodization aTR.
Yet, this is not sufficient, as the excitation region of each tilted plane wave has further be taken into account. A second apodization aPW is calculated depending on the tilting angle, the length and the distance from the ultrasound probe. aPW = 1, if the focus point is in direct range of the of the tilted plane wave. If it is out of direct radiation, a Gaussian windowing is performed depending on the angular difference. Furthermore, the Gaussian window is shaped depending on the distance to the ultrasound probe. A broader window is applied for higher distances, as the diffraction of the plane wave increases with the distance. The angular apodization is furthermore multiplied with a Heaviside function to set negative values to 0.
Finally, both apodizations are multiplied a = aTR * aPW

5) A clutter filter is already applied. This process is performed in two stages: In beamformed time signals and in the intensity values. With the first to filter coherent clutter by subtraction of the average time signal in each beamformed focus point. The second uses the B-Modes to subtract the clutter, which is done in the module PtvPy as “background removal” prior to the particle tracking. This acts like a long term spatio-temporal-filter. What is left-over, are supposedly distortions due to clipping, where the information loss cannot be recovered or the clutter not correctly calculated.
For clarification, we added this sentence right after the part of the apodization:
“Clutter filtering was applied by coherent subtraction of the average beamformed time signals and the background removal in the B-Mode data with PtvPy.”

6) First, the absolute vector of the two dimensional deviation is calculated with the root mean square. Second, the resulting absolute vector of the localization deviation is transformed to a localization uncertainty. This is done in accordance to the expression of uncertainty in measurement (GUM). The calibrated average localization deviation must be considered as an unknown systematic deviation. Therefore, its probability density distribution must be modelled as a worst case scenario, which is in this case an uniform distribution. The expression of a measurement uncertainty or rather a localization uncertainty requires a Gaussian distribution. Therefore, the uniform distribution can be translated to a Gaussian distribution by dividing it with the square root of 3.
As two steps are performed within one equation, we split it to the part of the absolute vector calculation and the uncertainty calculation. We further added a remark to GUM.

We thank you for your review and this constructive discussion! We hope we have addressed all your concerns.

References

[1]     Vogt, T., Ishimi, W., Yanagisawa, T., Tasaka, Y., Sakuraba, A. and Eckert, S., 2018. Transition between quasi-two-dimensional and three-dimensional Rayleigh-Bénard convection in a horizontal magnetic field. Physical Review Fluids, 3(1), p.013503.
[2]    Vogt, T., Yang, J.C., Schindler, F. and Eckert, S., 2021. Free-fall velocities and heat transport enhancement in liquid metal magneto-convection. Journal of Fluid Mechanics, 915.
[3]    Yang, J.C., Vogt, T. and Eckert, S., 2021. Transition from steady to oscillating convection rolls in Rayleigh-Bénard convection under the influence of a horizontal magnetic field. Physical Review Fluids, 6(2), p.023502.
[4]    Akashi, M., Yanagisawa, T., Sakuraba, A., Schindler, F., Horn, S., Vogt, T. and Eckert, S., 2022. Jump rope vortex flow in liquid metal Rayleigh–Bénard convection in a cuboid container of aspect ratio. Journal of Fluid Mechanics, 932.
[5]    Kupsch, C., Feierabend, L., Nauber, R., Büttner, L. and Czarske, J., 2020. Ultrasound super-resolution flow measurement of suspensions in narrow channels. IEEE Transactions on Ultrasonics, Ferroelectrics, and Frequency Control, 68(3), pp.807-817.
[6]    Weik, D., Nauber, R., Kupsch, C., Büttner, L. and Czarske, J., 2021. Uncertainty quantification of ultrasound image velocimetry for liquid metal flow mapping. IEEE Transactions on Instrumentation and Measurement, 70, pp.1-11.

Reviewer 2 Report

Weik et. al. presented their study of ultrasound localization microscopy application in liquid metal flow mapping.  The aim of this work is to enable flow mapping with high spatial resolution.  Rayleigh-Bénard convection experiment is applied to demonstrate ultrasound localization microscopy to study liquid metal flow characteristics.  To obtain high spatial resolution, an adaptive beam forming method is implemented. They demonstrated that a spatial resolution of 188 micrometers can be obtained which corresponds to 0.28 times the ultrasound wavelength. This approach was shown to enable observation of high velocity gradients within the recirculation areas of liquid gallium-indium-tin metal convection rolls. The paper is well-written, and the data presented supports the claims of the paper. 

I recommend acceptance of the manuscript in its current form. 

Author Response

Thank you for your review!

Reviewer 3 Report

The presented manuscript entitled "Ultrasound Localization Microscopy in Liquid Metal Flows" is an excellent experimental paper presenting a novel flow measurement method for the ultra-fine resolution especially of the steep gradient regions. The paper has a scientific soundness and can be recommended for the publication.

However, to complete this study a necessary modification can be expected: the detailed set of the boundary and initial conditions for the experiment setup in Figure 4 and velocity profiles plots for characteristic cross-sections (center-line of the cavity in horizontal direction, probe line crossing the centers of the small recirculation zones, etc.) of the measured flow field in Figure 5. That will allow this work to be a reference and benchmark case for the numerical studies, which other researches could perform, and, moreover, definitely will perform after this novel publication. Thank you!

Author Response

We added the suggested information and the velocity plots to the manuscript. We hope you can find now the data of which you are interested in. Thank you for your efforts, your review and your suggestions in order to strengthen our paper!